

# Sporting a virtual future: exploring sports and virtual reality patents using deep learning-based analysis

Jea Woog Lee[1], Sangmin Song[2], JungMin Yun[2], Doug Hyun Han[3] and YoungBin Kim[4]

[1] College of Sport Sciences, Chung-Ang University, Ansoeng, Republic of South Korea
[2] Department of Artificial Intelligence, Chung-Ang University, Seoul, Republic of South Korea
[3] Department of Psychiatry, Chung-Ang University Hospital, Seoul, Republic of South Korea
[4] Graduate School of Advanced Imaging Science, Multimedia and Film, Chung-Ang University, Seoul, Republic of South Korea

## ABSTRACT

We investigate the convergence of sports and emerging technologies from the Fourth Industrial Revolution, with a focus on virtual reality (VR) applications. Using patent big data, we introduce SportsBERT, a bidirectional encoder representation from transformers (BERT)-based algorithm tailored for enhanced natural language processing in sports-related knowledge-based documents. Through topic modeling, we extract key themes and clusters from sports-related VR patents, providing insights into the knowledge structure and technological trends in VR applications for sports. Our analysis identifies key drivers of technological advancement, including spatial hardware, tactile human–computer interactions, aerobic exercise, rehabilitation, and swing sports. Additionally, we highlight challenges such as the high cost and usability limitations of current VR devices. This study presents the first deep learning-based topic modeling approach specialized for sports patents and offers a comprehensive roadmap for current developments and future trajectories in VR sports technologies.

## INTRODUCTION

With the advent of the Fourth Industrial Revolution, the integration of sports and advanced technologies has become increasingly prevalent (*Mahmood & Mubarik, 2020*). This convergence aims to enhance efficiency and problem-solving capabilities within the sports domain, leading to the development of specialized technological innovations tailored for the sports industry. Fundamental technologies of the Fourth Industrial Revolution, such as virtual reality (VR), augmented reality (AR), artificial intelligence (AI), big data, wearable devices, and robotics, are being extensively utilized in both industry and academia to develop sports-specific applications (*Deng et al., 2022*). Among these, VR stands out as a transformative technology in the smart-device industry (*Fox, 2016*). VR enables the creation of highly immersive environments through rendered images

Corresponding author
YoungBin Kim, ybkim85@cau.ac.kr

or graphics, facilitating applications in training, education, and entertainment (*Villagran-Vizcarra et al., 2023*). Technological advancements in sports have further highlighted the significance of VR. Numerous sports competitions and major sporting events, including professional league tournaments, have adopted VR broadcasting services to enhance viewer engagement (*Rynarzewska, 2018*). Furthermore, national athletes benefit from scientifically optimized training programs that leverage VR-based mental training and simulation scenarios, allowing them to practice in realistic game environments while mitigating spatial and temporal constraints as well as injury risks (*Putranto et al., 2023*). The emergence of metaverse technology has further expanded the applications of VR in sports. During the COVID-19 pandemic, metaverse platforms facilitated virtual fan interactions and remote exercise programs, offering innovative solutions to engagement challenges (*Zhu et al., 2023*). The integration of VR technology within the sports industry has not only stimulated technological advancements but also served as a strategic tool in addressing the disruptions caused by the pandemic (*Kim & Ko, 2019*). However, while these examples illustrate the potential of VR in sports, many aspects of this technology remain underexplored. As technological innovation accelerates, predicting future developments becomes increasingly complex and uncertain (*Zimmerling & Chen, 2021*). Additionally, despite the evident promise of certain technologies, navigating the vast landscape of patents poses a significant challenge, particularly concerning VR applications in sports. In recent years, the categorization of emerging technologies has become increasingly ambiguous, as technological advancements often emerge through interdisciplinary convergence (*Aaldering, Leker & Song, 2019*). Consequently, the pursuit of scientific and technological progress requires a comprehensive approach that examines both macro- and micro-level developments. Patents offer valuable insights into the evolution of technology, with patent databases encompassing over 90% of documented technological advancements (*Willoughby & Mullina, 2021*). These repositories contain detailed technical descriptions of developed technologies, enabling the analysis of past trends and the projection of future innovations (*Jeong & Yoon, 2015*). Moreover, patent analysis helps mitigate technological redundancy, bridge knowledge gaps, and foster innovation by identifying untapped technological potential. In the context of sports, recognizing promising technologies is essential for shaping science and technology policies and formulating effective research and development (R&D) strategies (*Ratten, 2019*).

Patent documents, however, present a unique challenge for analysis. They are often lengthy and contain highly technical or newly coined terminology, making it difficult for conventional text-mining approaches to fully capture their meaning (*Haghighian Roudsari et al., 2022*). Traditional patent analysis methods, such as manual review or simple keyword and citation counts, may struggle to identify underlying themes and emerging trends, particularly in an interdisciplinary domain like sports VR. Innovations in sports VR frequently span multiple fields, further increasing the complexity. This underscores the need for advanced modeling techniques capable of understanding the nuanced language and context within patent documents.

To address this challenge, this study employs a deep learning approach based on Bidirectional Encoder Representations from Transformers (BERT). BERT is a state-of-the-art language model that has demonstrated effectiveness in capturing contextual relationships in text, even within highly technical domains (*Bonaccorsi, Melluso & Massucci, 2022*). By leveraging a pre-trained transformer model and fine-tuning it on patent data, this study aims to interpret the content of sports VR patents with greater accuracy and efficiency. This approach is particularly well-suited for patent analysis, as it can process complex sentence structures and domain-specific terminology, enabling the identification of subtle patterns that simpler models might overlook (*Zhu et al., 2022*). In essence, using BERT allows for a deeper and more precise analysis of patent data, aligning well with the intricacies of patent language and content, making it a powerful tool for this task. This study is based on a comprehensive patent dataset encompassing sports-related VR innovations. Specifically, patent records were compiled from major patent offices worldwide, covering a 32-year period (1994–2022). This broad temporal and geographical scope provides a comprehensive perspective on the evolution of VR applications in sports. By examining patent filings across different timeframes and jurisdictions, this study traces technological advancements and identifies key areas where sports VR innovation has gained traction. Establishing this scope is essential for contextualizing the findings and assessing their generalizability. Building on this foundation, the study aims to explore the intersection of sports and VR through patent analysis, employing state-of-the-art deep learning techniques. The key objectives are as follows: (1) to map the landscape of sports VR innovation by analyzing patent filings, highlighting major technological domains and developments in the field; (2) to demonstrate the effectiveness of a BERT-based deep learning model in processing complex patent text and extracting meaningful insights; and (3) to identify emerging themes and future trends in sports VR technology, as revealed by recent patent activity, to inform practitioners and guide further research.

As mentioned above, the application of VR into sports has facilitated technological convergence, with several notable applications. However, the expectation of quantifiable growth does not necessarily ensure qualitative advancements in future VR landscapes. Despite the rising interest in VR, its widespread adoption remains limited because of economic factors, particularly concerns regarding usability and high costs from the consumer perspective (*Laurell et al., 2019*). Nonetheless, VR holds significant potential to shape the next generation of sports and eSports, offering new avenues for participation in the post-pandemic era (*Peng et al., 2022*; *Westmattelmann et al., 2021*). These contrasting developments underscore the need for a comprehensive understanding of long-term technological strategies to facilitate the effective integration of VR into the sports industry.

The main contributions of this study can be summarized as follows.

1. **Development of SportsBERT: a domain-specific natural language processing (NLP) algorithm:** This study introduces a specialized adaptation of the BERT algorithm, termed SportsBERT, designed to enhance the interpretation of knowledge-based documents in the sports domain through advanced NLP techniques.

2. **Data-driven analysis of emerging sports technology trends:** By applying topic modeling to patents related to sports and virtual reality, this study identifies key themes,

major keyword clusters, and the evolving knowledge structure of VR applications in sports.

3. **Pioneering deep learning-based topic modeling for sports patents:** This research represents the first attempt to implement a deep learning-based topic modeling approach for analyzing specialized sports-related patents, providing valuable insights into technological advancements in the field.

Securing a technological edge not only strengthens market influence but also fosters sustained innovation within a given domain (*Bekhet & Latif, 2018*). The temporal relevance of sports patents—considering factors such as shared growth perspectives (*e.g.*, sports as a public commodity), the broad industrial applicability of patents, and the role of advanced technologies in shaping the industry—underscores the necessity of an initial analytical framework. The findings of this study are expected to provide practical insights into the future trajectory of VR technologies in the sports sector.

## THEORETICAL BACKGROUND

Topic modeling is a crucial technique in NLP for identifying common themes and underlying narratives within a text. It is primarily categorized into probabilistic and graphical methods (*Vázquez et al., 2022*). Over time, topic modeling has evolved from latent semantic analysis to its probabilistic counterpart, with latent Dirichlet allocation (LDA) emerging as the dominant approach. In recent years, deep learning-based methods have gained prominence, particularly BERTopic, a topic modeling framework leveraging a BERT autoencoder. BERTopic has become widely adopted because of its ability to capture contextual relationships in text. Additionally, the use of domain-specific models has been shown to significantly enhance topic modeling performance, ensuring more accurate and meaningful topic extraction.

LDA is a widely used topic modeling technique (*Blei, Ng & Jordan, 2003*). It generates two matrices: (1) topic-per-document and (2) word-per-topic, creating a distribution of distributions. This allows LDA to assess topic and word distributions across documents and generalize new data efficiently. However, traditional topic modeling methods like LDA are static (*Hida et al., 2018*) and assume that all topics are independent, neglecting topic correlations (*Blei, Ng & Jordan, 2003*). In reality, diverse text data, such as social media content or hierarchically structured information, exhibit interrelated topics (*Sangaraju et al., 2022*). Consequently, LDA struggles with large datasets and fails to accurately predict new documents. Additionally, LDA assumes that words near a cluster's centroid are the most representative of that cluster and its topic (*Yau et al., 2014*). However, clusters do not always conform to spherical shapes, leading to misrepresentation of certain topics.

Owing to the inconsistent performance of traditional topic modeling techniques, pre-trained language models have gained adoption. A prominent example is BERT (*Wang et al., 2023*), which has demonstrated significant success in topic extraction (*Scepanovic et al., 2023*; *Schadeberg et al., 2023*; *Wanchoo et al., 2023*). Its advanced language representation capabilities and bidirectional processing enable it to capture nuanced relationships between words and contexts, thereby improving the accuracy and quality of topic extraction.

Furthermore, BERT leverages large-scale pre-training on diverse text sources, facilitating comprehensive semantic interpretation and contextualization. These characteristics contribute to its effectiveness in topic modeling. In particular, BERTopic enhances topic extraction by incorporating clustering techniques and a class-based variant of term frequency-inverse document frequency (TF-IDF) to generate cohesive topic representations (*Grootendorst, 2020*). Unlike centroid-based approaches, BERTopic improves topic coherence by refining clusters through dynamic topic embedding and adaptive term weighting. Comparative studies have evaluated the performance of pre-trained models against traditional topic modeling techniques across various domains. For instance, BERTopic outperformed LDA and latent semantic analysis in analyzing the Consumer Financial Protection Bureau dataset, as well as LDA and non-negative matrix factorization in processing data from three Arabic newspapers (*Ogunleye et al., 2023*; *Sangaraju et al., 2022*). Additionally, a domain-specific model, FinBERT, was compared with BERTopic for financial text analysis (*Huang, Wang & Yang, 2023*). Owing to its fine-tuning on financial documents, FinBERT exhibited superior performance, yielding more distinct topics and higher coherence scores (C_V and U_Mass) than BERTopic. Despite extensive research on topic modeling in various fields, limited studies have explored its application in the sports domain, particularly in analyzing patent data.

## RELATED WORK

The application of topic modeling in sports research has expanded significantly over the past decade. Researchers have increasingly integrated advanced machine learning algorithms, including topic modeling techniques, into sports analytics to extract meaningful insights from diverse data sources such as social media posts, news articles, player statistics, and game records. These efforts have facilitated the identification of emerging trends and public sentiments surrounding major sporting events (*Hidayatullah et al., 2018*). Additionally, topic modeling has been employed to classify sports-related tweets and articles, enabling real-time sentiment analysis and research trend identification for specific teams, players, and events. Such applications have proven valuable in shaping team marketing strategies and enhancing fan engagement initiatives (*Zelenkov & Solntsev, 2023*).

Topic modeling has been utilized in sports analytics to examine play-by-play data, offering valuable insights into game strategies, pattern recognition, and trend forecasting in competitive sports (*Li et al., 2014*). For instance, analyses of National Basketball Association and European soccer matches have identified key game patterns, strategic formations, and player roles that are not readily captured by traditional statistical methods (*Principe et al., 2022*). While topic modeling has been widely applied to sports-related articles, news reports, and social media data, its application to patent analysis in sports remains an underexplored area with significant potential. Despite rapid technological advancements in sports, the absence of comprehensive patent analysis research has left substantial gaps in understanding the sports technology landscape. This lack of exploration hinders insights into innovation trends and emerging technologies. Consequently, addressing this research gap is crucial for both academia and industry. A structured approach to patent analysis

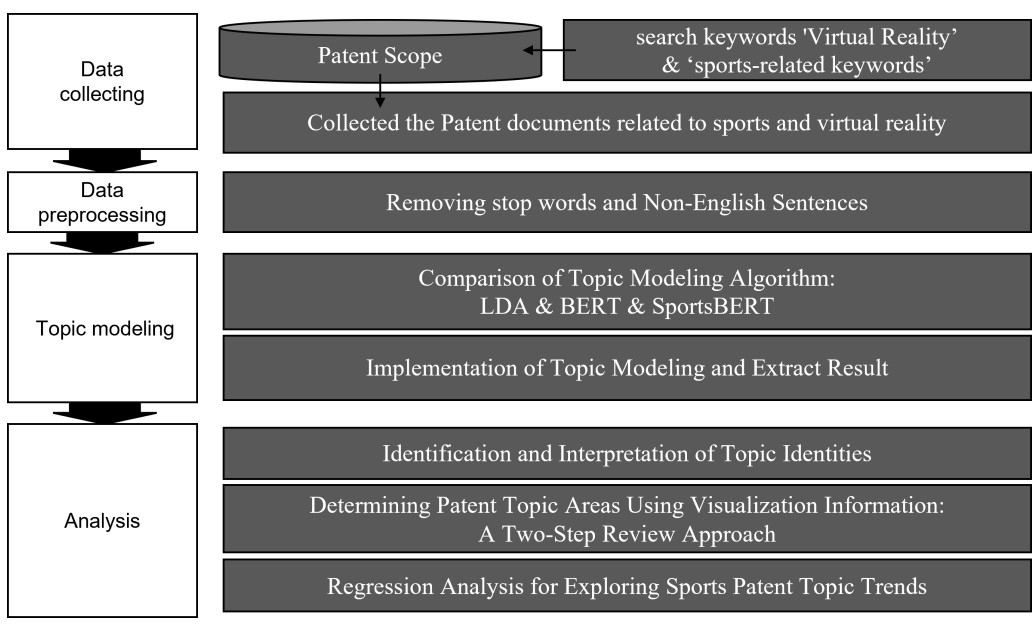

**Figure 1** Overall research flow.

can enhance the understanding of the technological evolution in sports, facilitating patent discovery, trend forecasting, and innovation assessment. In this study, topic modeling is applied to analyze VR-related patents in sports, establishing a foundational framework for future research on patent analytics in the sports industry.

## MATERIALS & METHODS

The methodology of this study is structured into several key steps, as summarized in Fig. 1. First, a corpus was created by collecting patent documents from PATENTSCOPE. The collected data were then assessed using coherence measures to ensure the suitability of topic modeling. Based on the evaluation results, the most effective topic modeling technique was selected. Subsequently, topic modeling was conducted using SportsBERT, a sports-specific model, to extract topic clusters. The keywords and patent documents corresponding to each topic were analyzed and classified to interpret the identified topics. Visualization tools were employed to examine topic distributions, while regression models were applied to analyze temporal trends in patent topics. This comprehensive approach enabled a deeper understanding of the current landscape and future prospects of sports-related VR technologies, providing insights into promising research directions.

### Data collection and dataset structure

Patent documents related to sports and VR were collected from PATENTSCOPE (https://patentscope.wipo.int). The search keywords included "virtual reality" and "virtual", combined with sports-related terms such as "sport", "exercise", "physical activity", and "fitness", along with all sports categories recognized by the International Olympic Committee (http://www.olympics.com). To ensure data reliability, only registered patents

that had obtained exclusive intellectual property rights—excluding abandoned, expired, or pending applications—were included in the analysis. As a result, a total of 19,388 patents related to sports and VR technologies were collected, covering a 32-year period from 1994 to 2022.

## Comparison and implementation of topic modeling

To identify the most suitable topic modeling technique for patent data, we compared traditional approaches, such as LDA, with a transformer-based model called BERTopic (*Grootendorst, 2020*). The evaluation was conducted using patent data obtained from the World Intellectual Property Organization (WIPO). Additionally, we examined whether domain-specific, fine-tuned embeddings, such as SportsBERT (*Huggingface, 2020*), could outperform other topic modeling techniques. This comparative analysis enabled us to determine the most effective approach for extracting meaningful insights from sports patent data. For performance evaluation, we utilized C_V and U_Mass coherence measures. These metrics provided a quantitative assessment of topic coherence and interpretability, reflecting each technique's ability to capture logically structured and meaningful themes in patent data. By comparing these measures, we aimed to identify the optimal topic modeling technique for patent analysis in the sports domain.

We implemented LDA for topic modeling using the Gensim library (version 4.3.1) to analyze text data. The data were processed using the Natural Language Toolkit (NLTK) (version 3.8.1) by applying tokenization, stop-word removal, and lemmatization using the WordNet lemmatizer. The preprocessed data were then transformed into a bag-of-words representation to create a dictionary and corpus. To determine the optimal number of topics, we iteratively trained LDA models with varying numbers of topics using the LdaModel class from Gensim. The C_V coherence measure was computed using the CoherenceModel class in Gensim to empirically select the most appropriate number of topics. As shown in Fig. 2, the maximum coherence score (C_V = 0.5687) was achieved at four topics. Therefore, we analyzed topic modeling using three topics to determine coherence (*Mimno et al., 2011*).

Additionally, we evaluated BERTopic, a BERT-based topic modeling technique, for comparison with LDA. Two variations of BERTopic were used:
1.  A general-purpose model (BERT-based uncased model).
2.  A domain-specific model (SportsBERT).

The text data underwent the same preprocessing steps, such as tokenization and stop-word removal using NLTK. Common generic words, such as "copyright", "invention", "figure", "VR", and "virtual", were frequently present but did not contribute significantly to the formation of semantically meaningful topics. Hence, they were added to the stop-word list. In BERTopic, document embeddings were generated using the corresponding BERT models. Topics were extracted through dimensionality reduction *via* uniform manifold approximation and projection (UMAP) and hierarchical clustering. Apart from the embedding model used to generate document embeddings, all topic modeling techniques were evaluated under the same conditions. The coherence scores of both the

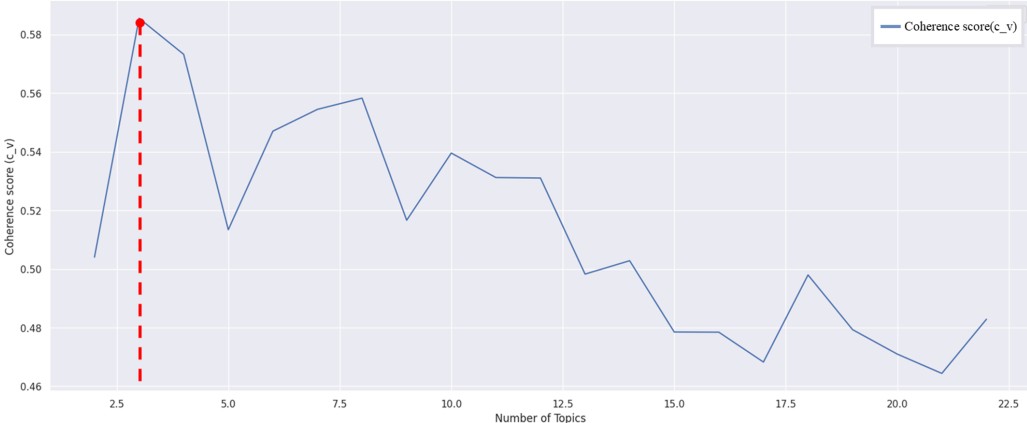

**Figure 2** Coherence score according to number of topics when using optimal LDA model.

**Table 1 Coherence scores for different topic modeling techniques.**

| Technique | Measure | |
|---|---|---|
| | C_V | U_Mass |
| LDA | 0.5687 | −1.5870 |
| BERT | 0.5991 | −2.0255 |
| SportsBERT | 0.6195 | −2.1072 |

general-purpose (BERT-based uncased model) and domain-specific (SportsBERT) models were compared.

The three topic modeling techniques were assessed based on coherence scores. Specifically:

1. C_V quantified the topic similarity.
2. U_Mass measured intra-topic word distances across the three models.

As shown in Table 1, BERTopic with SportsBERT demonstrated higher performance, achieving coherence scores of C_V = 0.6195 and U_Mass = −2.1072, indicating that it effectively captured meaningful and coherent sports-related topics. The topics extracted using SportsBERT provided valuable insights into the latent themes in the selected patent data, enabling a comprehensive analysis. Based on these results, topic modeling with SportsBERT was adopted in this study as a robust, domain-specific approach, enhancing our understanding of the underlying patterns and topics in sports-related text data. An optimal topic model was determined by minimizing intra-topic distance and maximizing inter-topic distance, ensuring well-separated and meaningful topic clusters.

## Model robustness and reliability analysis

Given the novel application of a BERT-based model to patent topic analysis, additional experiments were conducted to verify the robustness and reliability of the results. Inspired by previous BERT-based topic modeling research, we performed a series of sensitivity

analyses on the SportsBERT topic model to ensure that the findings were not artifacts of parameter choices or random initialization.

1. Topic stability across multiple runs

To assess topic stability, the embedding and clustering process was repeated multiple times using different random seeds. The results showed that core topics remained consistent across runs—the same thematic groupings of patents were observed, and the average topic coherence exhibited only marginal variations. This consistency indicates that SportsBERT's clustering results are not highly sensitive to random initialization.

2. Performance stability on data subsets

The dataset was split into two subsets based on publication year, and topic modeling was performed independently on each subset. The major topics identified in the full dataset were also present in both subsets, confirming that no single time period disproportionately influenced the results. Additionally, a leave-p-out analysis was conducted by randomly holding out a percentage of patents and rerunning the topic modeling to examine changes in topic structure. The held-out patents were then used as a test set, verifying that they could be correctly assigned to their closest matching topics from the remaining data. This analysis confirmed the generalizability of the identified topics.

3. Parameter sensitivity testing

To further confirm model reproducibility, key parameters were systematically varied, including:

- Number of clusters/topics (k)
- Dimensionality reduction settings

Results indicated that while extremely low or high values of k led to topic merging or over-splitting, a broad range of k values consistently produced the same dominant themes. Similarly, minor modifications in text preprocessing did not substantially alter the identified topics. These findings demonstrate that the SportsBERT-based topic modeling approach is robust—the topics and trends remain stable under various perturbations, confirming their reliability. All analyses were conducted in a computationally reproducible pipeline, and the code and processed dataset have been made available for verification by other researchers. This comprehensive robustness testing, combined with expert validation, ensures that the insights derived from the patent data are both credible and reproducible.

## Expert validation of identified topics

To ensure that the topics identified by our models are meaningful and aligned with domain knowledge, we conducted an expert validation study. Five domain experts with strong backgrounds in sports technology and VR were engaged to review and validate the discovered topics. These experts represent both academia and industry: two are university professors specializing in sports technology and computer science with a focus on VR applications, one is a senior patent analyst with over a decade of experience in sports and VR-related patents, one is a VR software developer in the sports training industry, and one is an R&D manager at a sports technology company. All experts hold advanced degrees

in relevant fields, including three Ph.D. degrees and two M.Sc. degrees, ensuring their qualifications in interpreting sports and VR innovations.

1. Evaluation criteria

To objectively assess the coherence and relevance of topics produced by LDA and SportsBERT, the experts were provided with a predefined set of evaluation criteria adapted from established patent topic modeling studies within the sports–VR domain. The criteria included the following:

Coherence—Assessing whether patents grouped under a given topic share a clear common theme. For instance, do the title and abstracts of patents in a topic relate to a specific concept such as VR-based athlete training? High coherence implies that the topic's top keywords and patents are semantically related and form a logical grouping.

Distinctiveness—Determining whether each topic is well differentiated from others. This ensures that topics are not redundant or significantly overlapping. For example, a topic on virtual sports coaching systems should be distinct from one on VR gameplay analytics.

Relevance—Evaluating whether the topic represents a meaningful and significant theme in the sports VR domain, rather than being an artifact of text processing or an insignificant cluster. This criterion checks if a topic is recognizable and useful for domain experts studying sports technology innovations.

2. Expert validation process

Each expert was provided with a summary of each topic and independently assessed it using the above criteria. They assigned qualitative ratings and flagged any unclear or inconsistent topics. Following the independent evaluations, a joint review meeting was conducted, where the experts discussed their findings, resolved discrepancies, and refined topic definitions.

As a result, a finalized set of validated topics was established, ensuring that they were coherent, distinct, and relevant. Additionally, expert feedback was used to assign descriptive labels to each confirmed topic. This expert validation step enhances the objectivity and credibility of the topic modeling results, ensuring that the reported topics provide real-world insights into the technological landscape of sports and VR.

## Visualization method for topic area determination in VR sports using a two-step review

Topic modeling provides diverse visual insights, enabling an understanding of similarities and associations between topics. Through visualization tools, we can explore the relationships between topics and assess clustering based on their relevance to sports patent documents. Each topic has distinct characteristics, and patent domains can be identified by analyzing the similarities and associations between topics. Sports encompass a wide range of disciplines, each involving specific equipment, movements, and technological advancements. However, multiple disciplines share similar biomechanical principles and technological applications. Therefore, topic modeling should reveal the application of VR technology across different sports and its unique characteristics. In this study, we employed

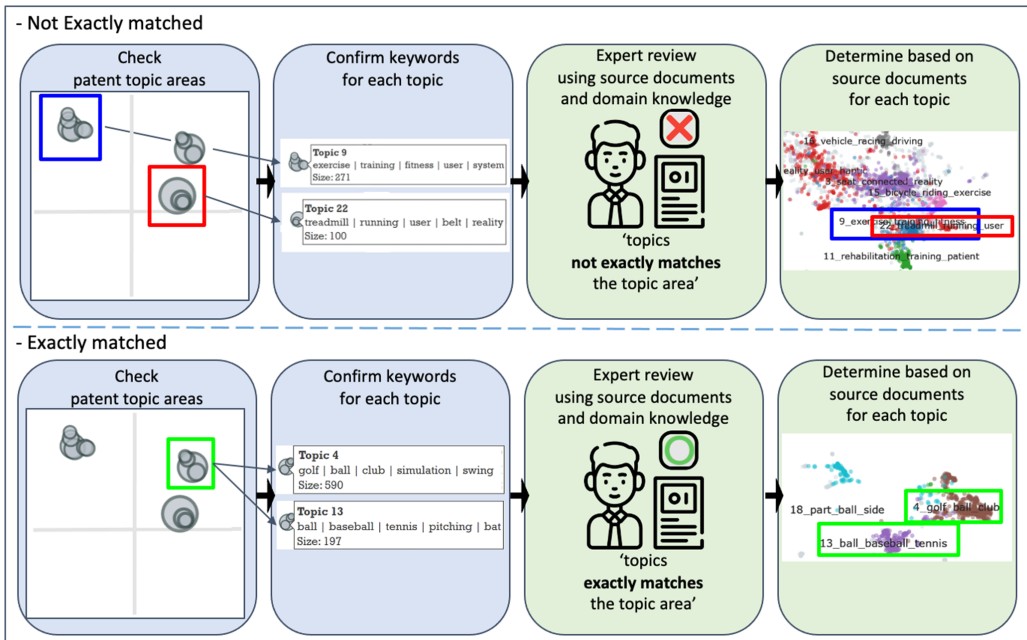

**Figure 3** Example of two-step review used to analyze visual information from VR technology applied to sports.

a two-step review process using various visualization tools to determine representative topic areas. As shown in Fig. 3, the process was structured as follows:

1. Step 1: Topic clustering *via* distance mapping (blue boxes)
   - A topic distance map was used to identify clusters of topics, forming distinct topic areas within sports patent data.
   - Keywords per topic were then organized into specific topic areas, ensuring that related themes were grouped appropriately.
2. Step 2: Expert review and refinement (green boxes)
   - Domain experts in sports technology examined the original patent documents to verify whether topics were correctly assigned within their respective areas.
   - If an expert review indicated that a topic did not fully align with its assigned area, the topic was reallocated to a closer relevant area in the document map, based on the embedding of the documents forming the topic.
   - The source documents were then analyzed using expert domain knowledge to confirm their placement.
   - If a topic was found to be independent, meaning it did not fit within any established area, it was categorized as a standalone topic.

This systematic visualization and validation process enabled a clear understanding of the convergence between sports and VR technologies, ensuring that topic modeling results were both structured and meaningful.

### Regression analysis to explore topical trends in sports patents

Technological development generally follows distinct stages of growth, maturity, and decline over time. Our analysis of available data indicates that VR technologies in sports have undergone various phases of change, growth, and decline since 1994. To examine these trends, we utilized the topic proportion metric, which was calculated by dividing the frequency of a topic in a specific year by its overall frequency derived from topic modeling results. This provided time-series data, enabling a quantitative analysis of topic evolution. We then applied regression models to analyze trends within topic areas, allowing us to:

1. Assess the current state of sports VR technology based on patents classified under each topic.
2. Infer future developments by identifying patterns of growth or decline in topic prevalence.

This approach provides a structured and data-driven perspective on the evolution of VR-related innovations in sports, offering valuable insights for researchers, industry professionals, and policymakers.

## RESULTS

### Results of topic modeling

To analyze the knowledge structure and trends in VR patents related to sports, we applied SportsBERT-based topic modeling to identify and categorize 23 distinct topics within the dataset. Each topic was grouped into clusters based on its thematic relevance to broader topic areas. Additionally, a two-step visual analysis was performed to ensure coherent topic identification and structuring. The topic modeling results are presented in Table 2, summarizing the 23 identified topics. During topic derivation, the domains of each topic were automatically labeled, reflecting the most significant conceptual linkages between the four most relevant keywords for each topic. To ensure accuracy, the topic identities were further refined through expert evaluation and comprehensive analysis of keywords, source documents, and patent metadata. Furthermore, a time-series regression analysis was conducted to explore temporal trends across the identified topics, providing insights into the evolution of VR applications in sports over time. The following section presents a detailed summary of each extracted topic, highlighting its thematic focus and relevance in the context of sports and VR innovations.

### Topic 0: VR-based game operations and interface control

This topic encompasses VR game operations and interface control mechanisms for sports applications. It includes patents related to virtual training and gaming systems that enable users to interact with and control elements within a virtual sports environment. Key components include control interfaces, scene generation, and interactive props, as identified in the extracted keywords. Representative patents include US 2016/0077547, titled "System and Method for Enhanced Training Using a Virtual Reality Environment and Bio-Signal Data" (*Aimone et al., 2018*), which describes a VR-based training system allowing users to engage in simulated sporting events through interactive elements. Another relevant patent, US 2006/0122035, "Virtual Reality Exercise System and Method" (*Felix,*

**Table 2   Topic modeling results (topics and keywords) obtained from VR patents applied to sports.**

| Topic 0 | Topic 1 | Topic 2 | Topic 3 |
|---|---|---|---|
| operation_scene_control_game | image_space_display_game | reality_user_haptic_device | seat_connected_reality_utility |
| operation | image | reality | seat |
| scene | space | user | connected |
| control | display | haptic | reality |
| game | game | device | utility |
| according | solution | system | base |
| comprises | solved | data | plate |
| interface | camera | environment | experience |
| prop | position | motion | body |
| medium | means | includes | end |
| steps | character | one | platform |
| Topic 4 | Topic 5 | Topic 6 | Topic 7 |
| golf_ball_club_simulation | sound_audio_source_voice | avatar_user_world_environment | game_currency_online_transaction |
| golf | sound | avatar | game |
| ball | audio | user | currency |
| club | source | world | online |
| simulation | voice | environment | transaction |
| swing | information | users | player |
| course | space | avatars | system |
| putting | output | one | user |
| face | means | client | items |
| unit | position | server | stock |
| sensing | signal | universe | item |
| Topic 8 | Topic 9 | Topic 10 | Topic 11 |
| augmented_AR_reality_real | exercise_training_fitness_user | game_server_terminal_player | rehabilitation_training_patient_limb |
| augmented | exercise | game | rehabilitation |
| AR | training | server | training |
| reality | fitness | terminal | patient |
| real | user | player | limb |
| system | system | means | upper |
| user | reality | machine | module |
| image | data | players | system |
| display | device | information | reality |
| device | module | match | hand |
| scene | unit | solution | robot |
| Topic 12 | Topic 13 | Topic 14 | Topic 15 |
| game_apparatus_player_character | ball_baseball_tennis_pitching | game_player_gaming_network | bicycle_riding_exercise_wheel |
| game | ball | game | bicycle |
| apparatus | baseball | player | riding |

**Table 2** (*continued*)

| player | tennis | gaming | exercise |
|---|---|---|---|
| character | pitching | network | wheel |
| space | bat | players | speed |
| image | sports | server | bike |
| unit | system | information | road |
| display | image | system | body |
| camera | screen | one | resistance |
| controller | pitcher | environment | rear |

| Topic 16 | Topic 17 | Topic 18 | Topic 19 |
|---|---|---|---|
| vehicle_racing_driving_car | information_processing _space_apparatus | part_ball_side_member | winning_combination _number_symbol |
| vehicle | information | part | winning |
| racing | processing | ball | combination |
| driving | space | side | number |
| car | apparatus | member | symbol |
| steering | unit | machine | symbols |
| control | position | surface | machine |
| road | device | board | reel |
| track | based | game | game |
| game | operation | body | reels |
| race | camera | outer | stop |

| Topic 20 | Topic 21 | Topic 22 |
|---|---|---|
| image_dimensional _three_display | ball_golf_surface_cover | treadmill_running_user_belt |
| image | ball | treadmill |
| dimensional | golf | running |
| three | surface | user |
| display | cover | belt |
| unit | core | reality |
| processing | spherical | support |
| space | diameter | walking |
| camera | hardness | motion |
| images | ratio | driving |
| moving | layer | rotary |

*2006*), focuses on VR-integrated sports equipment, demonstrating the integration of physical props within virtual sports experiences.

### Topic 1: Image display and spatial tracking in VR sports

This topic covers visual representation and spatial tracking in VR-based sports applications. It includes patents related to image rendering, motion capture, and the transformation of real-world actions into virtual simulations. The extracted keywords highlight key aspects such as VR-based display of sports imagery, user movement tracking *via* cameras, and character representation in virtual game spaces. A notable example is US 10,486,050, "Virtual Reality Sports Training Systems and Methods" (*Reilly et al., 2016*), which presents a VR-based system for visualizing player positions in a sports training environment.

Another patent, US 9,380,225, "Systems and Methods for Receiving Infrared Data with a Camera Designed to Detect Images Based on Visible Light" (*Tiscareno, Jonhson & Lawrence, 2016*), describes an infrared-based motion tracking system that captures user movements and replicates them in a virtual environment, enhancing the realism of sports training simulations.

### Topic 2: Haptic feedback devices for VR sports

This topic focuses on haptic feedback technology in VR sports applications, emphasizing devices that enhance immersion through tactile responses. Haptic feedback systems allow users to experience physical sensations that correspond to virtual interactions, improving the overall realism of VR-based sports training. Notable patents in this area include US 9,839,851, "Allowing Media and Gaming Environments to Effectively Interact and/or Affect Each Other" (*Ng & Lampell, 2017*), which introduces a wearable device that delivers haptic feedback synchronized with VR simulations. Additionally, US 10,425,577, "Image Processing Apparatus and Imaging Apparatus" (*Abe & Okamoto, 2019*), details a system that provides haptic responses based on user interactions with virtual sports equipment, such as balls and rackets, thereby enhancing the realism of sports-based VR experiences.

### Topic 3: Seated VR experiences in sports

This topic encompasses VR applications designed for seated sports experiences, focusing on the integration of ergonomic seating, motion platforms, and interactive hardware to enhance immersion. Patents in this category address adjustable seating systems that synchronize with VR environments, providing users with a dynamic and realistic experience. A key example is US 10,331,087, "Atom Interferometry in Dynamic Environments" (*Kotru et al., 2019*), which describes a seat that tilts and adjusts in response to user movements, creating a more immersive VR interaction. Similarly, US 10,430,270, "System for Migrating Data Using Dynamic Feedback" (*Chowdhury, 2019*), presents a seat-and-base system integrated into VR setups, enabling real-time movement detection and interaction within virtual sports simulations. These technologies facilitate greater physical engagement while ensuring comfort in seated VR experiences.

### Topic 4: VR-based golf simulation

This topic focuses on VR applications for golf training and simulation, emphasizing technologies that replicate real-world golf interactions through sensor-based swing analysis, virtual course rendering, and real-time feedback systems. Key patents include US 6,322,455, "Interactive Golf Driving Range Facility" (*Howey, 2001*), which describes an immersive VR golfing system that integrates physical club and ball sensors to provide real-time swing and trajectory feedback. Another significant patent, US 9,084,925, "Golf Swing Analysis Apparatus and Method" (*Davenport & Reynolds, 2015*), outlines a system that uses multiple sensors to track swing mechanics, offering detailed feedback to enhance player performance within a virtual training environment. These advancements highlight the role of VR in refining golf techniques and improving user engagement in digital sports training.

### Topic 5: Spatial audio and voice interaction in VR sports

This topic explores the role of sound in enhancing immersion within VR sports environments, with a focus on spatial audio rendering, voice-based interactions, and audio-driven user experiences. Patents in this category highlight innovations in 3D sound positioning and voice-controlled sports simulations. For instance, US 6,428,449, "Interactive Video System Responsive to Motion and Voice Command" (*Apseloff, 2002*), details a VR system that incorporates voice commands for interactive sports training and simulation. Another relevant patent, US 2022/0030375, "Efficient Spatially Heterogeneous Audio Elements for Virtual Reality" (*Falk et al., 2024*), describes advanced 3D spatial audio rendering techniques, enhancing the realism of VR-based sports applications. These developments demonstrate the importance of audio feedback in creating highly immersive VR sports environments.

### Topic 6: User avatars and multi-user VR sports environments

This topic explores the representation and control of user avatars in VR sports applications, encompassing avatar customization, multi-user interactions, and the underlying client–server architectures supporting these virtual environments. Notable patents in this category focus on enabling immersive user representation in shared VR spaces. For example, US 10,242,501, "Multi-User Virtual and Augmented Reality Tracking Systems" (*Pusch & Beall, 2019*), presents a VR system that allows multiple users to interact within a shared virtual environment, each represented by a customizable avatar. Similarly, US 9,526,983, "Virtual Reality Avatar Traveling Control System and Method" (*Lin, 2016*), describes a system for avatar creation, movement control, and personalization, which is particularly relevant for sports simulations and competitive virtual environments. These technologies facilitate the embodiment of users in VR sports, enabling diverse identity representations and immersive participation.

### Topic 7: Virtual economy and transactions in VR sports games

This topic examines the economic dimensions of VR sports, including in-game currencies, virtual transactions, and digital asset trading. Key patents highlight systems that support financial interactions within virtual sports environments. For instance, US 9,858,125, "System and Method for Optimizing Migration of Virtual Machines among Physical Machines" (*Bose & Sundarrajan, 2018*), details a transaction system for online games, which may extend to economic activities in VR sports gaming. Another relevant patent, US 11,383,169, "Systems and Methods for Adjusting Online Game Content and Access for Multiple Platforms" (*Wakeford & Huttula, 2022*), outlines a framework for earning, spending, and managing virtual currency within online sports games. These developments indicate that VR sports games not only serve as platforms for physical activity, exercise, and competition but also integrate economic mechanisms, fostering digital commerce within virtual environments.

### Topic 8: AR in sports applications

This topic investigates the role of AR in enhancing sports training and viewing experiences, highlighting AR-based performance analysis, real-time feedback systems, and AR-enhanced

sports broadcasts. A key patent, US 10,118,080, "Systems, Devices, and Methods for Virtual and Augmented Reality Sports Training" (*Loduca, 2018*), describes an AR system that provides athletes with real-time augmented visual feedback during training, improving situational awareness and decision-making. Another notable patent, US 11,140,378, "Sub-Picture-Based Processing Method of 360-Degree Video Data and Apparatus Therefor" (*Lee & Oh, 2021*), presents a system for overlaying AR graphics onto live sports broadcasts, enhancing the viewing experience for fans by providing real-time statistics, player information, and interactive elements. The overlap between AR and VR in sports technology underscores their shared technical foundations and complementary applications in sports performance analysis and fan engagement.

### Topic 9: VR applications in exercise, fitness, and sports training

This topic examines the integration of VR in exercise, fitness, and athletic training, focusing on systems that simulate training environments, track performance metrics, and enhance traditional workout routines. Patent US 10,549,153, "Virtual Reality and Mixed Reality Enhanced Elliptical Exercise Trainer" (*Fung, 2020*), describes a VR-enabled elliptical trainer that immerses users in diverse virtual environments, creating an engaging workout experience. Similarly, patent US 8,861,091, "System and Method for Tracking and Assessing Movement Skills in Multidimensional Space" (*French & Ferguson, 2014*), outlines a VR system that monitors user performance, provides real-time feedback, and tracks progress over time. This topic highlights a direct application of VR in sports, emphasizing its role in fitness and performance training.

### Topic 10: Server–client architecture in multiplayer VR sports games

This topic addresses the technical infrastructure supporting multiplayer VR sports games, including server–client architecture, player matchmaking, and in-game communication. Patent US 8,968,098, "Method and Apparatus for Online Gaming on Terminals" (*Harari, 2015*), details a system for connecting multiple players in a VR sports game *via* a centralized server. Likewise, patent US 10,569,163, "Server and Method for Providing Interaction in Virtual Reality Multiplayer Board Game" (*Hsueh et al., 2020*), presents a matchmaking system that optimizes player selection based on predefined parameters, enhancing the multiplayer gaming experience in VR. This topic underscores the critical role of networking technologies in enabling interactive and competitive VR sports environments.

### Topic 11: VR-based rehabilitation and motor function recovery

This topic explores the application of VR in sports rehabilitation, particularly for upper limb recovery and motor skill enhancement. Patent US 17/550744, "Virtual Reality Therapy System and Methods of Making and Using Same" (*Wheelbarger, Eskew & Matthews, 2022*), describes a VR-based rehabilitation system designed to restore motor function in patients, including injured athletes. Additionally, patent US 8,706,241, "System for Patient-Interactive Neural Stimulation with Robotic Facilitation of Limb Movement" (*Firlik et al., 2014*), details a system combining VR with robotic assistance to deliver personalized rehabilitation training, particularly for upper limb mobility in sports-related recovery. This

topic encompasses innovative VR systems aimed at improving physical therapy, motor coordination, and rehabilitation outcomes in sports medicine.

### Topic 12: VR hardware and user interface in sports games

This topic examines the hardware components and user interface mechanisms in VR sports games, including controllers, spatial representations, character control systems, and visual display technologies. Patent US 8,616,946, "Game Device, Control Method for Game Device, and Information Storage Medium" (*Yanagisawa & Ishida, 2013*), details a specialized hardware setup for VR sports games, incorporating controllers, cameras, and display units. Similarly, patent US 10,039,980, "Video Game Processing Apparatus and Video Game Processing Program Product" (*Matsui & Akiyama, 2018*), outlines a system for controlling in-game characters, manipulating game spaces, and interacting with virtual objects. This topic reflects the integration of immersive VR technologies in various sporting simulations, enabling smooth user interaction and enhancing the realism of VR sports environments.

### Topic 13: VR applications in ball sports

This topic focuses on the use of VR in ball sports, particularly baseball and tennis, emphasizing the simulation of pitching, hitting, and swinging movements in virtual environments. Patent US 11,040,287, "Experience-Oriented Virtual Baseball Game Apparatus and Virtual Baseball Game Control Method Using the Same" (*Kang, 2021*), describes a VR baseball simulation that enables players to practice pitching and batting while receiving real-time feedback. Likewise, patent US 7,771,295, "Off-Court Tennis" (*Braun & Wynne, 2010*), presents a VR system designed for tennis training, allowing users to practice swings, serves, and other fundamental skills in a controlled setting. This topic highlights the extensive exploration of VR in sports where precise movement mechanics, such as swinging, are fundamental, with applications extending to both gaming and professional training.

### Topic 14: Networking and multiplayer systems in VR sports games

This topic investigates the networked aspects of VR sports gaming, including online multiplayer functionality, server–client architectures, and player data-sharing systems. Patent US 11,192,032, "System and Method for Session Management in a Multiplayer Network Gaming Environment" (*Degarmo & Schmitz, 2021*), describes a VR gaming system that enables multiple players to interact in a shared virtual space *via* networked server infrastructure. Similarly, patent US 11,097,193, "Methods and Systems for Increasing Player Engagement in Multiplayer Gaming Environments" (*Paquet, 2021*), introduces a system for storing and sharing player data, facilitating a more immersive and interactive VR sports experience. This topic underscores the technological advancements that support online participation in VR sports, reinforcing the collaborative and competitive nature inherent to many sports disciplines.

### Topic 15: VR applications in cycling and exercise training

This topic explores the integration of VR into cycling for exercise and training, encompassing systems that simulate cycling environments, track riding performance, and

adjust resistance to replicate real-world conditions. Patent US 12/188143, "Cardio-Fitness Station with Virtual-Reality Capability" (*Fisher, Thompson & Nicoli, 2010*), describes an exercise bike that employs VR to create immersive cycling experiences, enhancing user engagement. Similarly, patent US 11,266,872, "Virtual Road Surface Implementation-Type Bicycle Simulator" (*Lee, 2022*), outlines a VR cycling system that dynamically adjusts bike resistance based on virtual terrain, mimicking real-world cycling conditions. This topic highlights the role of VR in professional training, fitness participation, and cycling-based VR sports simulations, expanding accessibility and engagement in virtual cycling experiences.

### Topic 16: VR in motor racing and driving simulations

This topic focuses on the application of VR in motorsports, particularly racing simulations, covering vehicle control mechanisms, race track simulations, and immersive driving environments. Patent US 10,357,715, "Racing Simulation" (*Buxton & Alejandro, 2019*), details a VR-based racing simulator that enhances realism by incorporating steering control, track conditions, and competitive race scenarios. Additionally, patent US 8,298,845, "Motion Platform Video Game Racing and Flight Simulator" (*Childress, 2012*), describes a motion-based VR simulation system that replicates race car handling, offering an immersive driving experience. This topic underscores VR's role in motorsport training and virtual racing games, advancing both entertainment and professional skill development.

### Topic 17: Information processing and spatial tracking in VR sports

This topic examines data collection, processing, and spatial tracking technologies within VR sports systems, focusing on positional tracking, movement analysis, and real-time rendering techniques. Patent US 8,092,301, "Information Aggregation Games" (*Alderucci, Plott & Miller, 2012*), presents a system for collecting and analyzing player movements, interactions, and positional data in VR sports environments. Likewise, patent US 11,278,811, "Systems and Methods of Rendering Screen Effects for Movement of Objects in a Virtual Area" (*Yoshihara et al., 2012*), describes a camera-based tracking system that enhances the precision of player movement representation in VR sports simulations. This topic highlights the significance of advanced data processing in optimizing VR sports training, enhancing realism, and improving interactive gaming experiences.

### Topic 18: Interaction of virtual objects in VR sports

This topic explores the interaction between players and virtual sports equipment in VR simulations, covering balls, boards, and other physical elements within sports environments. Patent US 10,715,759, "Athletic Activity Heads-Up Display Systems and Methods" (*Dibenedetto et al., 2020*), describes a system that simulates realistic player interactions with various sports objects, including balls and boards. Additionally, patent US 16/276937, "Ball-Striking Assist Method, Ball-Striking Assist System, and Portable Electronic Device" (*Li, 2019*), outlines a system that enhances the physics simulation of sports equipment and player movements in a VR setting. This topic highlights the advancement of hardware technologies designed to enhance realism in ball-based VR sports experiences.

### Topic 19: VR sports betting and game mechanics

This topic focuses on the integration of gaming and sports simulations, particularly in VR-based sports betting systems. It includes mechanisms for generating winning combinations, virtual reels, and wagering interfaces in sports-themed gaming environments. Patent US 13/290762, "Gaming System, Method, and Device Including a Symbol-Changing or Augmenting Feature" (*Lange & Villagran, 2018*), describes a VR gaming system incorporating sports-related symbols and mechanics to enhance the betting experience. Similarly, patent US 10,304,278, "System, Method, and Apparatus for Virtual Reality Gaming with Selectable Viewpoints and Context-Sensitive Wager Interfaces" (*Lyons & Steil, 2019*), presents a VR sports betting system featuring dynamic viewpoints and interactive wager placements. This topic underscores the technological advancements contributing to the evolution of VR in the sports gaming and betting industry.

### Topic 20: 3D image processing and display in VR sports

This topic explores technologies for capturing, processing, and rendering 3D images in VR sports simulations, enhancing visual realism and immersive experiences. Patent US 10,410,418, "Computer-Readable Non-Transitory Storage Medium Having Stored Therein Information Processing Program, Information Processing System, Information Processing Apparatus, and Information Processing Method for Controlling Movement of a Virtual Camera in a Game Space" (*Kiuchi, Shikata & Mouri, 2019*), details algorithms and hardware implementations for capturing and processing 3D images in VR sports environments. Additionally, patent US 9,805,490, "Method for Scripting Inter-Scene Transitions" (*Oh, Schoonmaker & Chang, 2017*), outlines a VR system employing advanced display technology to render realistic 3D visuals in virtual sports settings. This topic highlights ongoing advancements in graphical fidelity, crucial for enhancing immersion and user experience in VR sports.

### Topic 21: Virtual golf ball simulation in VR sports

This topic explores the design and simulation of virtual golf ball properties in VR sports environments, focusing on surface texture, core hardness, diameter, and layering to achieve realistic ball behavior. Patent US 8,926,443, "Virtual Golf Simulation Device, System Including the Same, and Terminal Device, and Method for Virtual Golf Simulation" (*Woo & Ok, 2015*), describes a system that replicates golf ball properties and physics to enhance realism in VR golf simulations. Additionally, patent US 14/540043, "Virtual Golf Simulation Apparatus and Method" (*Jang, 2015*), presents a detailed physics-based modeling approach for simulating golf ball behavior in a VR golf environment. This topic emphasizes the importance of realistic ball physics in VR sports, catering to both professional athletes and general sports enthusiasts.

### Topic 22: VR-integrated treadmill systems for running and walking simulations

This topic focuses on the integration of treadmills and related exercise equipment into VR systems, enabling realistic locomotion experiences through motion tracking and virtual environment synchronization. Patent US 10,445,932, "Running Exercise Equipment with

Associated Virtual Reality Interaction Method and Non-Volatile Storage Media" (*Chen & Gu, 2019*), describes a VR treadmill system designed to create an immersive running or walking experience by integrating real-time motion tracking with virtual environments. Similarly, patent US 7,780,573, "Omni-Directional Treadmill with Applications" (*Carmein, 2010*), introduces a system that captures user movements on a treadmill and accurately translates them into VR spaces. This topic highlights technological advancements aimed at enhancing VR-based fitness experiences through precise movement tracking and motion recognition.

## Topic areas based on keyword analysis

The topic areas and distances between topics were primarily derived from the inter-topic distance map, as shown in Fig. 4, which illustrates the spatial relationships among the 23 extracted topics. In this map, each circle represents a topic, and the proximity or overlap of circles indicates thematic similarity. The model considers word distributions, keyword frequencies, and their relative importance to determine inter-topic relationships. A well-structured topic model typically exhibits substantial overlap among circles, reflecting cohesive thematic connections, whereas a poor model results in scattered, isolated topics. In this study, highly related topics were clustered, while distinct topic groups were positioned farther apart in the quadrants.

Topics 0, 1, 12, 17, and 20 overlapped, sharing keywords such as "game", "camera", "image", and "space". Although fewer topics overlapped, additional common keywords, including "character" and "apparatus", were also observed. This cluster was tentatively identified as a category related to hardware and software development for VR sports games. However, the topic area did not entirely conform to this interpretation, as some topics exhibited common keywords but belonged to different conceptual domains. Additionally, Topic 0, which encompasses general VR technology, required further evaluation for clearer categorization. By leveraging expert insights in VR sports technologies and applying additional visualization algorithms, we refined the topic clusters to enhance interpretability and accuracy.

Topics 2, 5, 8, 9, 11, and 16 shared keywords such as "system", "reality", and "user". However, this grouping did not fully align with a single theme, highlighting the challenge of assuming that all topics within a cluster are inherently related. Topics 9 and 11 were clearly linked to physical exercise in VR, sharing keywords like "exercise", "rehabilitation", "fitness", and "training". In contrast, Topic 16, which focused on racing sports, lacked a direct connection to physical exercise, requiring further evaluation. Other topics in this cluster were associated with haptic feedback, sound technologies, and AR, which were identified as distinct themes rather than a cohesive category. Topics 4, 13, 18, and 21 shared the keyword "ball", with some topics also including "golf" and "surface". This cluster appropriately grouped sports-related VR patents, particularly those involving swinging movements in sports like golf, baseball, and tennis. The alignment of keywords supported the identification of a well-defined VR sports category. Topics 3, 15, and 22 exhibited partial keyword overlap, including "body" and "reality". However, this cluster did not reveal a strong thematic identity, as no clear relationships between the topics were identified.

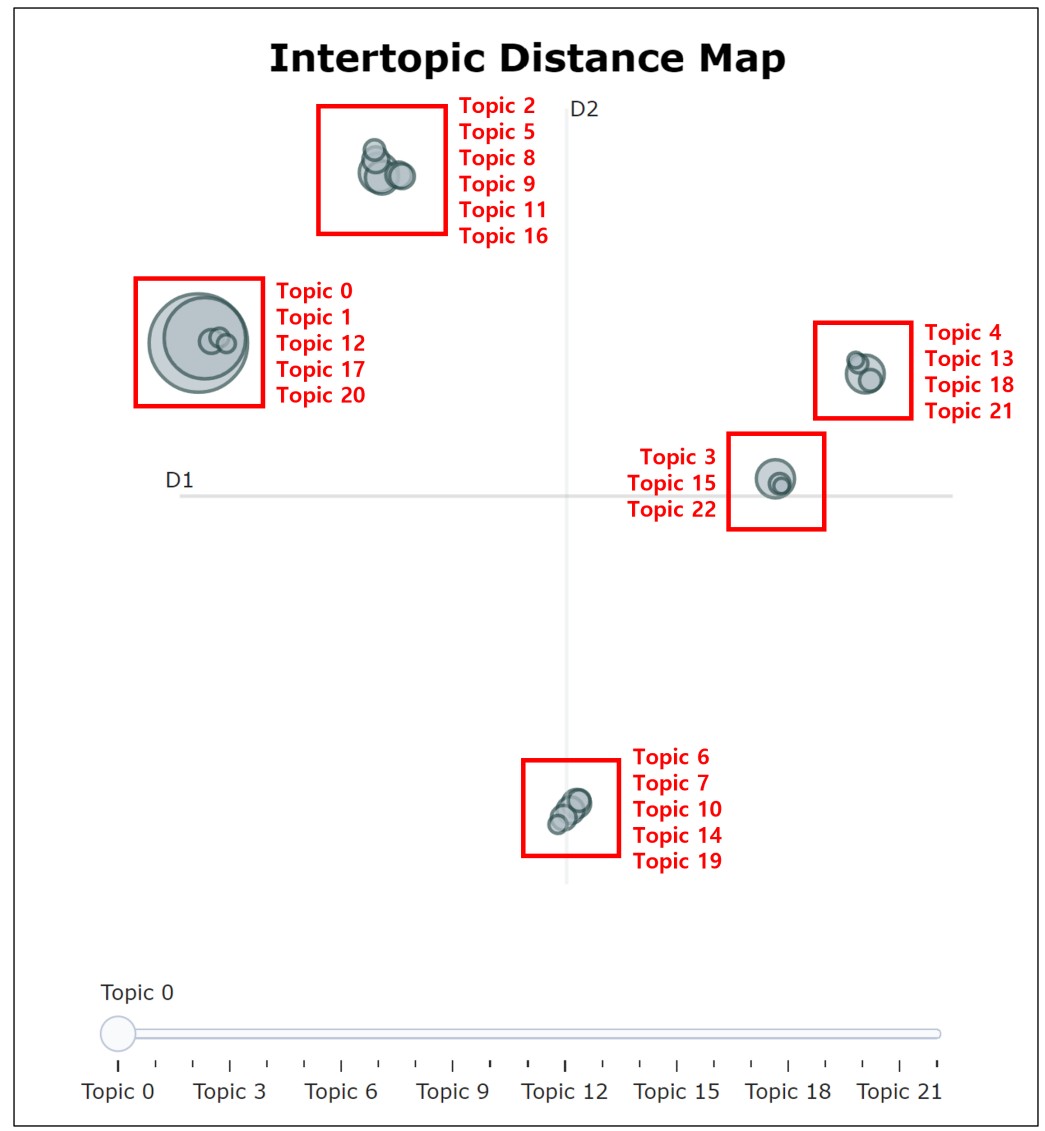

**Figure 4** SportsBERT topic modeling clusters.

Nonetheless, Topics 15 and 22 were closely linked through aerobic exercise applications in VR, such as cycling, running, and walking, while Topic 3 was associated with seating, platforms, and immersive VR integration. Given these distinctions, a clearer classification was required using expertise in VR sports technologies and advanced visualization tools. Topics 6, 7, 10, 14, and 19 shared keywords such as "game" and "player". These topics were inferred to be related to fundamental technical aspects of VR sports games. While most topics in this cluster aligned well, Topic 19 required further examination using keyword organization, source documents, visualization tools, and expert domain knowledge to ensure accurate classification.

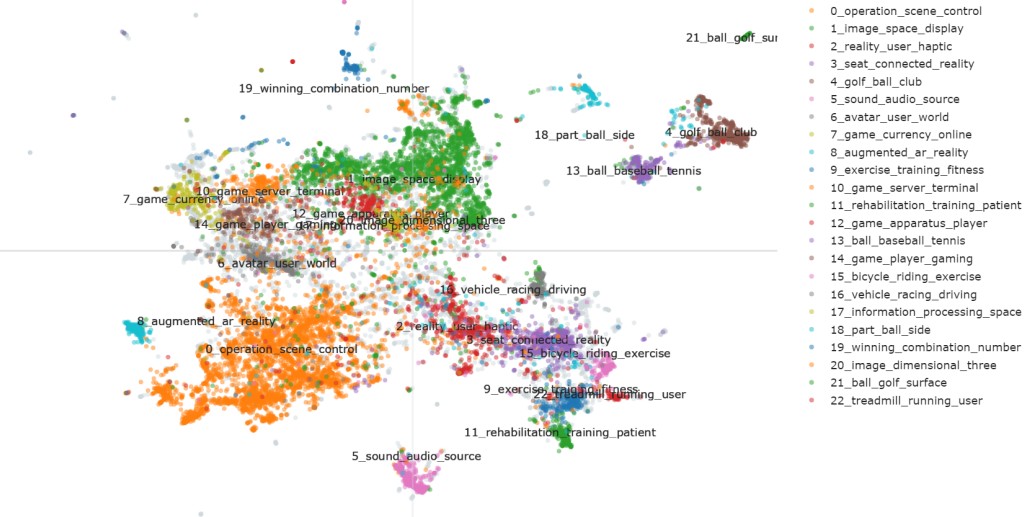

**Figure 5**  Document and topic maps activated for all nodes.

The inter-topic distance map is a clustering approach that visualizes the relationships between topics based on keyword commonality. However, some shared keywords identified by the model may not be directly relevant to a given topic. The map also revealed rare but partially low-relevance topics within clustered inter-topic relations. While topic clusters were generally well-separated across all quadrants, certain clusters appeared ambiguous and required reorganization.

## Topic areas based on patent documents

To better understand topic relationships, we used the inter-topic distance map for visualization. However, some topics needed to be reassigned to more appropriate areas. To address this, we employed a more detailed visualization approach, analyzing the distribution of documents within each quadrant to determine their correct topic areas. This approach involved recalculating document embeddings and projecting them into a two-dimensional (2D) space for improved visualization. Additionally, the semantic structure of sentences and phrases within each topic was analyzed to position topics more accurately. Since each document was represented as a node, topics with higher frequencies occupied larger areas in the quadrant. While it was challenging to define precise topic boundaries visually, as seen in the inter-topic distance map, the document-based clustering approach allowed for a more domain-relevant classification of VR sports topics. Figure 5 illustrates the document map, displaying nodes representing all topics. Users could refine the visualization by activating or deactivating topic domains through the right-column controls.

Figure 6 presents the maps for active Topics 2, 3, 9, 11, 15, 16, and 22. While these topics were separated into two clusters in the inter-topic distance map, the document map indicated stronger interrelations, with the topics positioned in close proximity. Notably, Topic 3 (seat_connected_reality_utility) was centrally positioned among the active topics, surrounded by document nodes related to the other topics. This topic comprises patents

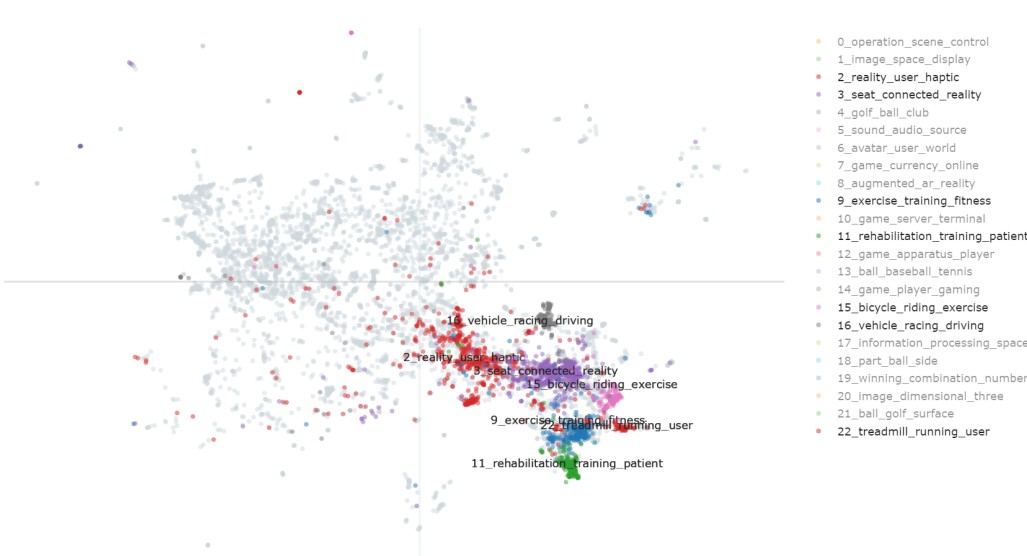

**Figure 6** Document and topic maps related to sports, exercise, and physical activity.

on physical plates, platforms, and seats designed for VR experiences, ensuring safe and effective integration with head-mounted displays (HMDs). Engaging in cardio or fitness exercises, such as biking or running in a VR environment, requires equipment that supports these movements while interacting with an HMD to reflect user motion. The proximity of Topic 3's document nodes to those of other topics suggests strong associations between these patented technologies. A similar pattern was observed for Topic 16, where document nodes, though separated by small gaps, clustered closely together. A realistic VR riding experience depends on an appropriate driving environment equipped with specialized hardware, leading to technical overlaps in patent relevance and clustering within the document map. Additionally, Topic 11 (rehabilitation) was positioned adjacent to Topic 9 (exercise training fitness user). While their document nodes did not directly overlap, topic 9 focused on technological advancements in VR-based fitness, which aligns with safety considerations and rehabilitation applications in Topic 11. Expanding the scope, Topic 2 also showed technical relevance to these areas. For users engaged in physical activity within a VR environment, tactile feedback, in addition to visual input, is essential for enhancing realism and immersion. Accordingly, the purpose, necessity, expected impact, and application strategies outlined in the patent documents suggest strong interconnections among VR developments in exercise, rehabilitation, and physical activity.

Figure 7 presents a visual representation of Topics 4, 13, 18, and 21. The document nodes formed a weak cluster, with considerable spacing between topics within a single quadrant. Despite their shared association with sports involving swinging movements, such as golf, baseball, and tennis, the distance between topic nodes suggests substantial differences in the underlying patent technologies. Each topic contained keywords related to movement, equipment, and materials associated with these sports. Topic 4 focused on sensor and measurement technologies for implementing golf simulators, while Topic 18

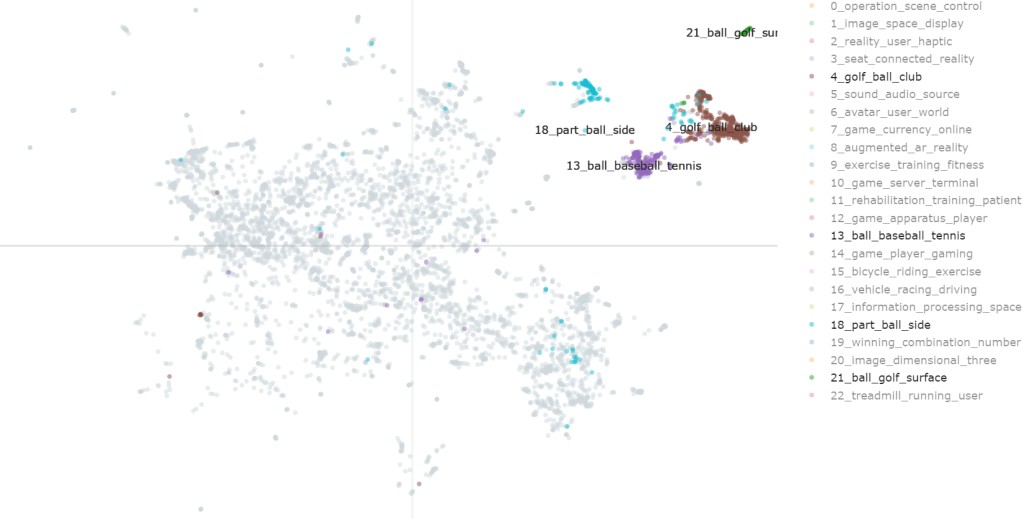

**Figure 7** Document and topic maps related to swinging sports.

emphasized physical interactions between users and VR devices. The proximity and partial overlap of nodes between these topics suggest a close technical relationship. Additionally, Topic 14 addressed specialized movement training for specific sports, though it also incorporated varied gaming objectives. Meanwhile, Topic 21 contained patents concerning realistic ball movement simulations in golf VR environments. This topic area, therefore, reflects an interesting interplay between shared elements and technological divergence across VR sports applications. Figure 8 illustrates the document map for Topics 6, 7, 10, and 14. Although Topic 19 was clustered in the inter-topic distance map, it was excluded from this visualization due to its greater distance from the other topics. The remaining topics remained closely clustered, all focusing on online VR sports experiences. The infrastructure of online VR environments includes servers, terminals, and interactive elements such as avatars, online transactions, and multiplayer gaming. As a result, this topic area primarily encompasses patented technologies that apply general VR advancements to sports applications.

Figure 9 presents document maps for Topics 1, 12, 17, and 20. Unlike the inter-topic distance map, where these topics were closely clustered with Topic 0 (operation_scene_control_game), the document map reveals an increased distance between them. The newly organized clusters are centered around visual technologies for VR, incorporating keywords such as "display" and "image", and hardware technologies, associated with terms like "apparatus" and "camera". Visual stimulation is one of the most intuitive experiences for users engaging in VR sports. This significance is reflected in the large area occupied by Topic 1 (image_space_display_game). The proximity of Topics 12, 17, and 20 to Topic 1 reinforces the shared thematic identity of this topic area. Figure 10 illustrates document maps for Topics 0, 5, 8, and 19. Topic 0 contains a broad range of patents conceptualizing VR devices and software in general, including applications for
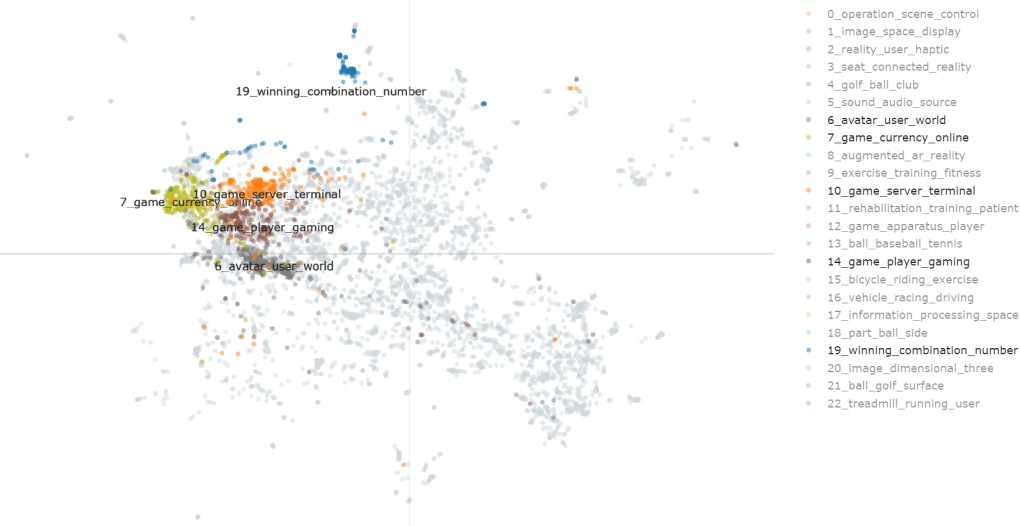

**Figure 8   Document and topic maps related to online environments.**

sports and VR environments. Since many of its underlying technologies are macroscopic in nature, they can be integrated into patents from other topics. This results in notable clustering in one quadrant, with some individual patent nodes distributed separately. Topic 5 features numerous patent nodes related to sound generation in VR devices and software. While it also includes patents related to display–gesture interaction, the document map indicates that patents focusing on sound signals may represent a distinct technological category. Topic 8 has a high concentration of patent nodes, focusing on the development of AR technologies. Although AR shares technical aspects with VR, it is fundamentally different, as it overlays digital information onto the real world rather than creating a fully virtual environment. This distinction suggests that patent nodes in this topic form an independent category. Topic 19, despite sharing the keyword ''VR sports games'' with other topics, primarily addresses betting within VR sports environments. Since playing sports in VR and gambling involve entirely different technological principles, the patents in this topic support the independent categorization of this area.

## Topic trends

Figure 11 presents the frequency of each topic over time. To assess the number of patent applications per topic, a linear regression analysis was conducted, using the year index as the predictor variable and the proportion of topics in that year as the dependent variable (*Jelodar et al., 2019*; *Paek, Um & Kim, 2021*). The study period began in 2010 due to the limited number of patents in earlier years, often zero or negligible. Focusing on this relatively recent period was deemed appropriate for regression analysis to examine trend patterns (*Resnik et al., 2015*). After estimating the slopes of the linear regression models,

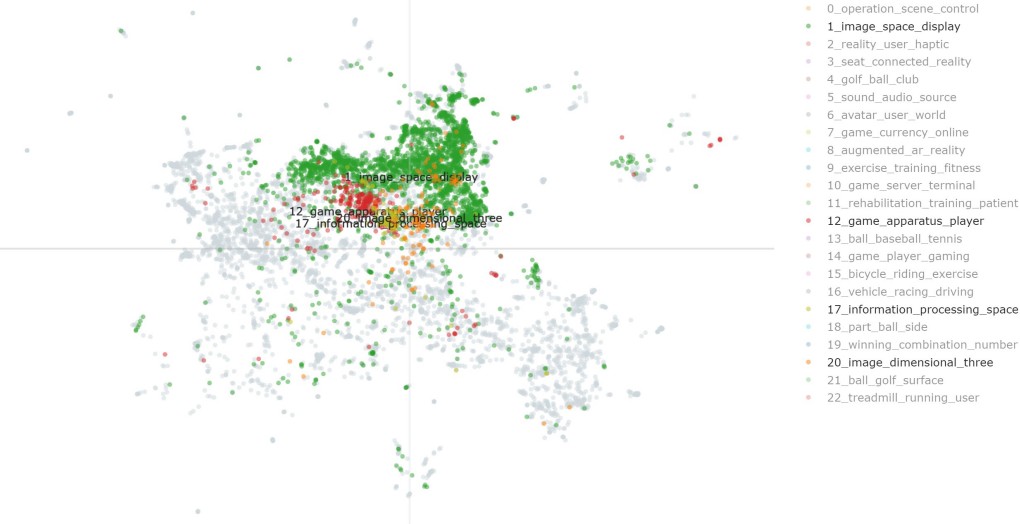

**Figure 9** Document and topic maps related to visual technologies.

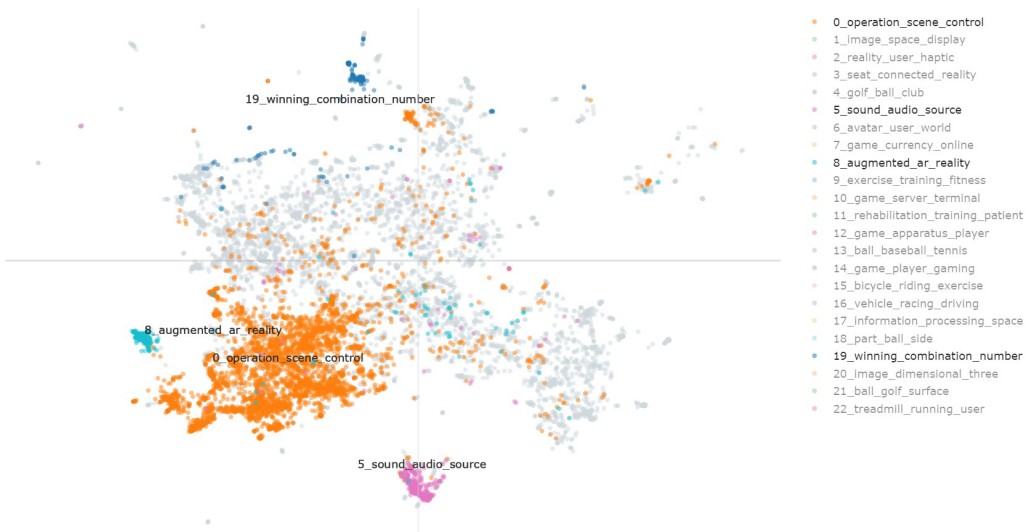

**Figure 10** Document and topic maps for independent topics.

the 23 topics were classified into four development categories based on their regression coefficients and statistical significance, as detailed in Table 3:

1. Hot topics: Topics with a steady increase in relative frequency over time (significant positive linear trend, $p$-value $< 0.05$).
2. Cold topics: Topics with a steady decline in relative frequency (significant negative linear trend, $p$-value $< 0.05$).
3. Warm topics: Topics with an increasing trend, but without statistical significance (positive linearity, non-significant $p$-value).
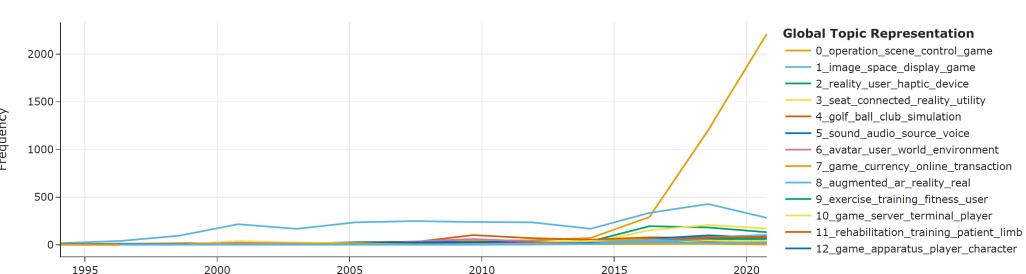

**Figure 11** Trends of studied topics over time.

**Table 3** Topic modeling results for patents related to VR sports.

| Topic area | Topic number | Topic domain | Coefficient | *t*-value | *p*-value | Category |
|---|---|---|---|---|---|---|
| Sports, exercise, and physical activity | 2 | reality_user_haptic_device | 0.732 | 3.566 | <0.01 | Hot |
| | 3 | seat_connected_reality_utility | 0.773 | 4.010 | <0.01 | Hot |
| | 9 | exercise_training_fitness_user | 0.847 | 5.278 | <0.001 | Hot |
| | 11 | rehabilitation_training_patient_limb | 0.476 | 1.797 | 0.100 | Warm |
| | 15 | bicycle_riding_exercise_wheel | −0.242 | −0.829 | 0.425 | Cool |
| | 16 | vehicle_racing_driving_car | 0.722 | 3.463 | <0.01 | Hot |
| | 22 | treadmill_running_user_belt | 0.807 | 4.538 | <0.001 | Hot |
| Swinging sports | 4 | golf_ball_club_simulation | 0.343 | 1.211 | 0.251 | Warm |
| | 13 | ball_baseball_tennis_pitching | −0.039 | −0.128 | 0.901 | Cool |
| | 18 | part_ball_side_member | 0.610 | 2.553 | <0.05 | Hot |
| | 21 | ball_golf_surface_cover | 0.032 | 0.107 | 0.917 | Warm |
| Online environment | 6 | avatar_user_world_environment | −0.226 | −0.768 | 0.459 | Cool |
| | 7 | game_currency_online_transaction | 0.181 | 0.612 | 0.553 | Warm |
| | 10 | game_server_terminal_player | 0.708 | 3.327 | <0.01 | Hot |
| | 14 | game_player_gaming_network | 0.834 | 5.021 | <0.001 | Hot |
| Visual technology | 1 | image_space_display_game | 0.392 | 1.414 | 0.185 | Warm |
| | 12 | game_apparatus_player_character | 0.594 | 2.451 | <0.05 | Hot |
| | 17 | information_processing_space_apparatus | 0.878 | 6.078 | <0.001 | Hot |
| | 20 | image_dimensional_three_display | 0.614 | 2.581 | <0.05 | Hot |
| Topic 0 | | operation_scene_control_game | 0.851 | 5.373 | <0.001 | Hot |
| Topic 5 | | sound_audio_source_voice | 0.471 | 1.770 | 0.104 | Warm |
| Topic 8 | | augmented_AR_reality_real | 0.683 | 3.099 | <0.05 | Hot |
| Topic 19 | | winning_combination_number_symbol | 0.447 | 1.657 | 0.126 | Warm |

4. Cool topics: Topics with a declining trend, but without statistical significance (negative linearity, non-significant *p*-value).

Key findings:

1. "Sports, exercise, and physical activity" topics were generally hot or warm, except for Topic 15 (bicycle_riding_exercise_wheel), which exhibited a weak declining trend.

2. "Swinging sports" topics (*e.g.*, golf, baseball, tennis) were mostly hot or warm, with the exception of Topic 13 (ball_baseball_tennis_pitching), which showed a slight downward trend.

3. The "online environment" area featured predominantly hot or warm topics, except for Topic 6 (avatar_user_world_environment), which was categorized as cool due to a weak decline.

4. "Visual technology" topics demonstrated hot or warm trends across the board.

5. Among the independent topics, Topics 0, 5, and 8 showed hot or warm trends, whereas Topic 19 exhibited a cool trend, reflecting a weak decline.

## DISCUSSION

We investigated the technological foundations shaping novel VR experiences in sports, a key innovation within the Fourth Industrial Revolution. Our evaluation, based on an analysis of VR patents, aimed to identify the driving forces behind advancements in VR-related sports technologies. Patents provide an intuitive perspective on technological developments; therefore, we employed SportsBERT topic modeling, a large-scale NLP method, to extract topics and topic domains. Additionally, we assessed trends for each topic using a regression model. Among the identified clusters, sports, exercise, and physical activity constituted the largest topic area. Patents related to Topics 2 and 3 highlighted spatial hardware technologies and tactile human–computer interactions as key contributors to ongoing technological advancements. Meanwhile, patents from Topics 9, 11, 16, and 22 provided a foundation for VR-based aerobic exercises such as running and cycling, as well as rehabilitation exercises, physical activities, and riding-based gaming experiences. The regression model projected sustained technological growth in these areas, with the exception of Topic 15, which exhibited a slight downward trend. This suggests potential challenges in predicting either continued development or decline. One possible explanation for this trend is the high cost of VR devices, which may limit their adoption compared to more affordable conventional sports equipment, such as bicycles (*Fromm et al., 2021*). Nevertheless, continuous advancements are integrating VR riding elements into fitness, training, and gaming applications (*Juras et al., 2019*). Furthermore, recent studies indicate a positive market outlook for VR-based cycling sports (*Kojić et al., 2023*). Overall, the steady progression of patent development in sports, exercise, rehabilitation, and riding technologies, alongside their applications in health maintenance, performance enhancement, and athletic training, reflects strong industry demand. Given the enduring need for aerobic exercise solutions, future research should explore emerging trends in patented VR technologies to ensure their continued evolution and integration.

The "swinging sports" topic area highlighted the integration of stick-based swinging movements into VR sports environments, particularly in ball sports such as golf, baseball, and tennis. Among these, VR golf has emerged as a dominant market in the United States, European Union, and Korea, where strong competition exists among manufacturers of 2D-based golf simulation screens (*Luczak et al., 2022*).

Patent analysis revealed that Topics 4, 18, and 21—which focus on VR golf technologies—accounted for 153 patent applications, the highest among VR sports

patents. VR golf integrates realistic course simulations with VR and sensor technologies to measure swing mechanics and ball trajectories, displaying the results *via* HMDs and 2D screens (*Hartfiel & Stark, 2021*; *Wang & Reid, 2011*). The golf simulation industry is well established in the USA, Europe, and Korea, with Korea's strong demand for screen golf accelerating industrial and technological advancements (*Han & Sa, 2022*; *Seong & Hong, 2022a*; *Seong & Hong, 2022b*). This growth is reflected in market projections, estimating the golf simulator market at USD 1,315 million in 2021, with an expected increase to USD 3,388 million by 2030, reflecting an annual growth rate of 10% (*Straits Research, 2022*). This suggests that patented VR golf technologies will continue to evolve in parallel with industry growth. In contrast, VR baseball exhibited a weak declining trend. The screen baseball market stagnated during the COVID-19 pandemic, reversing its prior growth (*Seong & Hong, 2022a*; *Seong & Hong, 2022b*). Despite similarities between screen baseball and screen golf, VR baseball simulations have been reported to feel less realistic, which may have contributed to lower engagement (*Seong & Hong, 2022a*; *Seong & Hong, 2022b*). Additionally, the keyword "tennis" in Topic 13 was challenging to differentiate from "baseball" in trend analysis. While tennis-related VR development remains in its early stages, recent virtual training intervention studies indicate an increasing application of VR technologies in tennis (*Le Noury et al., 2021*). For both baseball and tennis, market conditions, consumer demand, and technological advancements will influence future patent trends. Overall, golf, baseball, and tennis represent promising early applications of VR in swinging sports, signaling further potential for the convergence of sports and VR technologies.

The "online environments" topic area highlights technological advancements in software and hardware that facilitate measurements, analyses, and interactions throughout data transmission processes. The potential of these technologies was assessed through regression models, which emphasized their role in enhancing the realism of VR sports experiences (*He, 2022*; *Le Noury et al., 2021*; *Masai et al., 2022*).

Patent topic analysis revealed a focus on user–device interactions *via* wireless technologies, enabling greater freedom of movement, precise motion recognition, and hands-free operation without controllers. Accurate gesture measurement and interpretation are critical to ensuring seamless user interactions, directly influencing the quality of VR applications (*Ge et al., 2017*). In particular, VR-based training relies heavily on motion recognition technologies, as realistic body movements are essential for creating an immersive sports environment. A lightweight HMD, ergonomic fit, and wireless, controller-free design are key factors in maximizing user comfort and mobility in VR sports. Given the high risk of injuries in certain sports disciplines, VR training technologies are increasingly being explored for risk minimization and injury prevention (*Aida, Chau & Dunn, 2018*; *Maggio et al., 2019*). The "visual technology" topic area clustered patented advancements in 3D and 360° screens, HMD displays, 2D screens, and sports installations in virtual environments. This suggests a continued technological focus on creating versatile, immersive VR sports settings that are not restricted to specific disciplines. While screen-based VR sports experiences—such as screen golf and baseball—have traditionally relied on 2D flat-screen interactions, the increasing presence of "screen" as a keyword in this

topic area reflects the ongoing convergence of physical infrastructure and graphics-based gaming platforms (*Roettl & Terlutter, 2018*). Thus, HMD-based immersive experiences and flat-screen VR technologies are not mutually exclusive but rather complementary components of VR sports (*Hirota et al., 2019*). Regression models further predict sustained technological progress toward realistic visual experiences in VR sports, supported by an increase in patent clusters related to early VR sports technologies dating back to 2010. Although fluctuations in patent trends have been observed, the continuous growth and accumulation of patents indicate rising interest and investment in this domain. In the future, visual technology is expected to play a transformative role in the evolution of sports, while sports applications are likely to drive future advancements in VR technology. The high academic and socio-temporal value of VR sports technologies underscores the importance of continuous innovation and refinement in this field.

Our SportsBERT-based deep learning topic modeling approach leveraged a fine-tuned pre-trained BERT model to analyze sports VR patent documents, capturing domain-specific language and contextual nuances. This domain adaptation enabled the model to generate rich semantic embeddings—contextual vector representations of patent text—allowing for a deeper understanding of latent topic structures. By clustering these high-dimensional embeddings, we identified distinct technological themes, confirming the model's effectiveness in extracting coherent topic groupings from the dataset. However, despite its strengths, this BERT-driven topic modeling approach presents certain challenges. Patent terminology is often ambiguous or inconsistently applied, potentially leading to overlapping topic boundaries or redundant clusters. Additionally, the computational demands of the BERT pipeline are significant, particularly when processing large-scale patent corpora. Nonetheless, the flexibility of the BERT architecture ensures that our approach is not restricted to sports patents. The same fine-tuning methodology could be adapted to other patent domains, including healthcare, semiconductor technology, and green technology, to automatically uncover meaningful topics. This highlights the broad applicability of our approach across diverse technological fields, making it a valuable tool for large-scale patent analysis and innovation trend discovery.

## CONCLUSION

Patent analysis in sports remains an underexplored research area, yet it presents significant challenges and opportunities. Sports are increasingly intertwined with scientific and technological advancements, extending beyond traditional physical activities to encompass innovation in equipment, training, and wellness industries. Given these paradigm shifts, proactive analysis of VR sports patents and early engagement with emerging technologies are crucial for advancing sports science and technology. The proposed SportsBERT model, specifically adapted to the sports domain, provides meaningful insights into VR-related patent trends and serves as a valuable tool for future analyses. Understanding the impact of patent analysis in sports and its future directions is essential. Continued exploration and practical applications of this research will contribute to the evolution of sports technology.

The source code is available at GitHub repository: https://github.com/ezzy4me/wipo_topicmodel.git.

### Funding

This work was supported by the Institute of Information & Communications Technology Planning & Evaluation (IITP) grant funded by the Korea government (MSIT) (RS-2021-II211341, Artificial Intelligence Graduate School Program (Chung-Ang University)). This research was funded by the Chung-Ang University Research Grants in 2022. The funders had no role in study design, data collection and analysis, decision to publish, or preparation of the manuscript.

### Grant Disclosures

The following grant information was disclosed by the authors:
Institute for Information & Communications Technology Planning & Evaluation (IITP) grant funded by the Korea government (MSIT): RS-2021-II211341.
Chung-Ang University Research Grants in 2022.

### Competing Interests

The authors declare there are no competing interests.

### Author Contributions

- Jea Woog Lee conceived and designed the experiments, analyzed the data, prepared figures and/or tables, authored or reviewed drafts of the article, and approved the final draft.
- Sangmin Song analyzed the data, prepared figures and/or tables, and approved the final draft.
- JungMin Yun performed the experiments, authored or reviewed drafts of the article, and approved the final draft.
- Doug Hyun Han performed the experiments, performed the computation work, prepared figures and/or tables, and approved the final draft.
- YoungBin Kim conceived and designed the experiments, performed the computation work, prepared figures and/or tables, authored or reviewed drafts of the article, and approved the final draft.

### Data Availability

    The code and data is available at Github and Zenodo:
    - https://github.com/ezzy4me/wipo_topicmodel
    - sangmin song. (2025). ezzy4me/wipo_topicmodel: Initial Release for Zenodo DOI (v1.0.0). Zenodo. https://doi.org/10.5281/zenodo.15275142.

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
