# Peer review of "Sporting a virtual future: exploring sports and virtual reality patents using deep learning-based analysis"

_PeerJ Computer Science, doi:10.7717/peerj-cs.2919_

## Round 0.1 · original submission · Minor Revisions

Thanks for your submission. Kindly resubmit your revision addressing the reviewer's comments. Further, there are some typos and language-related issues, please correct them.

**Language Note:** The Academic Editor has identified that the English language must be improved. PeerJ can provide language editing services - please contact us at [email protected] for pricing (be sure to provide your manuscript number and title). Alternatively, you should make your own arrangements to improve the language quality and provide details in your response letter. – PeerJ Staff

·

Basic reporting

1. The introduction effectively explains the background and importance of studying VR technology. However, there still seems to be a gap between patent data modeling and the use of Deep Learning BERT. It would be beneficial to illustrate the complexities of patent data and explain why Deep Learning BERT is a suitable approach for processing such data.

2. The repetition of the research objectives on lines 81 and 88 is somewhat confusing. It would be clearer to present the research objectives at the end of the introduction after discussing all the issues and urgency, ensuring a logical flow before concluding with a clear statement of the research goals.

3. Adding information about the scope of the patent data being processed, such as geographical region, country, or time period, either in the abstract or introduction, would provide valuable context for the reader.

Experimental design

4. After collecting the patent data, it is important to explain the structure of the dataset and provide examples so that readers can better understand the format of the data processed by the LDA/BERT model. This structure may include elements such as the title, content, or specific paragraphs.

5. In the expert review section, the educational background and qualifications of the experts involved should be described. Mention the number of experts involved and clarify the validation process—did the experts establish specific criteria? Including these criteria in the manuscript would improve objectivity.

Validity of the findings

6. The discussion section provides a solid analysis of the research findings, but the effectiveness of the proposed SportBERT model has not been explored in depth. It would be helpful to elaborate on how topic modeling is conducted using deep learning, what challenges arise, and how BERT can be applied to other fields in patent data processing.

Reviewer 2 ·

Basic reporting

The article presents a study on the convergence of sports and advanced technologies, specifically virtual reality (VR), using patent data and deep learning analysis to assess the evolution of VR technologies applied to sports. In this context, an advanced natural language processing model, "SportsBERT," is employed to analyze documents related to VR sports patents. The article is well-structured and adequately explains the objectives, methodology, results, and conclusions.

Experimental design

The experimental design is robust in terms of the methodology used for patent analysis through natural language processing, specifically with a BERT-based model (Bidirectional Encoder Representations from Transformers). The utilization of "SportsBERT," a model specifically adapted for sports patents, is innovative and well-suited for this type of analysis.

Validity of the findings

The validity of the findings is influenced by the novelty of the methodology and the approach used in patent analysis, yet the lack of technical details regarding the validation of the models and results limits the ability to fully assess their validity. A more comprehensive sensitivity analysis or robustness testing for the SportsBERT model is missing, which could raise concerns regarding the reliability of the findings. It would be crucial for the authors to provide more details on how they ensured the consistency and reliability of the results.

Additional comments

The article could benefit from greater clarity regarding the technical implementation of SportsBERT and a more detailed explanation of the patent analysis process. A deeper discussion of the challenges mentioned, such as the high costs and low usability of current VR devices, would be valuable for the scientific community and could open new avenues for further research.

---

## Round 0.2 · accepted · Accept

The reviewer comments are well addressed. Congratulations.

·

Basic reporting

The author has accommodated input and revised the manuscript well.

Experimental design

The author has accommodated input and revised the manuscript well.

Validity of the findings

The author has accommodated input and revised the manuscript well.